# Influence of Social Exclusion on Cool and Hot Inhibitory Control in Chinese College Students

**DOI:** 10.3390/ijerph20032433

**Published:** 2023-01-30

**Authors:** Suhao Peng, Xinhui Ruan, Dan Tao, Bin Xuan

**Affiliations:** 1Department of Psychology, School of Education, Anhui Normal University, Wuhu 241000, China; 2School of Early Childhood Education, Nanjing Xiaozhuang University, Nanjing 211171, China

**Keywords:** social exclusion, ostracism, Cyberball, Stroop task, cool inhibitory control, hot inhibitory control

## Abstract

Social exclusion can affect nearly every aspect of a person’s mental health, both on an emotional and cognitive level. The purpose of the present study was to investigate whether cool or hot inhibitory control capacity varied under social exclusion. More precisely, participants who had experienced and not experienced social exclusion were compared to explore the influence of social exclusion on cool and hot inhibitory controls. Social exclusion was induced through the use of a Cyberball game, and participants were divided into an exclusion group and an inclusion group. The number Stroop task and emotional face Stroop task were used to measure the cool and hot inhibitory control, respectively. In the cool Stroop task, participants had to refrain from reading printed digits to identify the number of items presented in the array. In the hot Stroop task, participants had to inhibit the meaning of the word to identify the emotion displayed on the face. Reaction time, accuracy, and Stroop interference were analyzed to compare the inhibitory control between the exclusion group and the inclusion group. The results showed an extension of the response time in the exclusion group compared to the inclusion group. We found a higher interference effect in the number of Stroop tasks in the exclusion group than that in the inclusion group, but it was not significant in the emotional face Stroop task. The results suggest that the cognitive and emotional basis of inhibitory control may differ during social exclusion. The present findings expand our understanding of how social exclusion affects cool and hot inhibitory controls and their internal psychological mechanism.

## 1. Introduction

Human beings are naturally social animals and have a strong need for stable social relationships in groups. This need has an evolutionary underpinning, in which humans evolved to forge and maintain social connections with others to keep in physical and mental health [1,2]. However, social exclusion is a highly painful experience and poses a serious threat to these fundamental human needs [3,4]. Social exclusion is a pervasive phenomenon and process associated with a hostile living environment and can take many forms, such as rejection, refusal, isolation, and ignorance [5]. The rights or resources of an individual or group in an environment of social exclusion, such as employment, medical care, housing, political participation, etc., are not normally enjoyed because of racial discrimination, physical and mental discrimination, etc. [6,7]. Social exclusion has been associated with a variety of severe impairments across social, emotional, and cognitive domains [8,9]. According to the need-threat model, social exclusion threatens four basic needs, including a decrease in belonging, self-esteem, sense of control, and meaningful existence, compared to increases in anxiety, depression, aggressive and impulsive behaviors [10,11]. In relation to cognitive ability, social exclusion leads to the deconstruction of emotional control, behavior control, and memory bias, which is associated with risky, drunk driving, eating binges, spending sprees, alcohol and illicit drug abuse, violence, and many types of criminal behaviors [2,12]. Considerable research effort has been directed at understanding the emotional and cognition factors that are responsible for the negative outcome of social exclusion. In the laboratory, social exclusion has been induced through diverse manipulations, including ostracism, rejection, discrimination, and written material manipulations of social exclusion [13]. A consistent finding is a detrimental effect of social exclusion on the performance in emotional and cognitive processing at an early stage, which was dubbed the “reflexive and reflective stage” in the need-threat model [10]. However, some studies found that social exclusion leads to an increased ability to detect conflict in cognitive tasks, suggesting that exclusion does not always have detrimental effects on performance [14]. Therefore, the aim of the present study is to investigate how emotional and cognitive control varied under social exclusion.

Since both emotional turmoil and cognitive deconstruction are the main factors that underlie the pattern of behavior observed after social exclusion, research is needed to further explore the interaction between emotional and cognitive processing during social exclusion. Here, we investigated one appropriate candidate to measure the interaction, namely the executive function (EF), which has three factors in measurement, including inhibitory control, working memory, and cognitive flexibility. Zelazo and Carlson have divided EF into two aspects: hot EF and cool EF [15,16]. Even though both hot and cool EF are top-down processes that depend on cognitive effort, they require a variety of mental resources to manage emotional processing. Hot EF is involved in situations that are highly motivating and emotional, whereas cool EF is more prominent in situations that are not as emotional [15]. Recently, to better distinguish the two types of EF, a series of studies have specifically measured a hot and cool inhibitory control by using affective versus non-affective Stroop tasks, respectively [17,18,19,20]. Inhibitory control is a core executive function, which involves the ability to suppress automatic action, thought, or feelings to help reduce conflicting situations [21]. Indeed, the hot and cool inhibitory control relies on different processes and thus would be affected differently by specific social contexts. For instance, one study reported that social evaluation facilitated the hot inhibitory control in adolescents, but this facilitation effect was not found in the cool inhibitory control [22]. Another recent study suggested that cool and hot inhibitory control performance was not correlated in adolescents [23]. Taken together, these findings suggest that the hot and cool inhibitory control depends on different cognitive and emotional processing, which are similar to hot and cool EF.

It is possible that emotional arousal should be considered to measure the influence of the social context effect on inhibitory control, such as social exclusion [24]. There is growing evidence that an individual’s cognition, emotion, and behaviors are sensitive to social exclusion [25,26,27], but studies on how social exclusion affects inhibitory control still have many controversies. For example, the temporal need threat model suggests that social exclusion threatens the need for belonging and impairs the inhibitory control of impulsive behavior [4,11]. Cognitive deconstruction theory suggests that individuals who experienced social exclusion would enter a defensive state of cognitive deconstruction, such as slower reaction times and lower conflict control [28]. However, the social monitoring system found that social exclusion altered an individual’s attention to social cues, such as facial expressions [29,30]. The sociometer theory emphasizes that social exclusion increases the awareness of the relational value and social information to adjust self-esteem [31,32]. The key to the difference between these views is whether social exclusion promotes or inhibits cognitive processing related to social information [33,34]. Taken together, these theories and models of social exclusion suggest that inhibitory control was an important cognitive ability closely related to the increasing or decreasing behavior performances of social exclusion.

In addition, evidence has shown that social exclusion may interact with a variety of cognitive processes, including emotional regulation and inhibitory control [23,35,36]. It is likely that the behavior performance of excluded individuals is conditional on the reaction to the association between emotion and cognition during social exclusion. Since emotion and cognition responses induced by social exclusion are complex, inhibitory control can be affected differently by an affective and non-affective stimuli. Thus, it is valuable to investigate how social exclusion modulates affective and non-affective inhibitory control. This study aimed to examine the effects of social exclusion on cool and hot inhibitory control by using Cyberball combined with Stroop tasks. Following the above theories and models of social exclusion, we hypothesized that social exclusion facilitates hot inhibitory control in the emotional Stroop task but impairs cool inhibitory control in the non-emotional Stroop task.

## 2. Materials and Methods

### 2.1. Participants

Sixty-four undergraduate students (32 males; age = 21.3 ± 2.2 years) from Anhui Normal University were recruited in the present study. All the participants were right-handed, had normal or corrected-to-normal vision, and had no history of neurological problems. All participants were paid 20 RMB for their participation and gave written informed consent, which was approved by the Ethics Committee of Anhui Normal University, China.

Participants were divided randomly into two groups: the exclusion group (32 participants, 16 males) and the inclusion group (32 participants, 16 males). The two groups of subjects were ensured to achieve a homogeneous state by controlling self-esteem, anxiety, and depression. Three participants (one male and two females) were excluded from analyses due to the failure of ostracism manipulation and poor data quality.

### 2.2. Procedure

The experimental procedure was divided into five parts: (1) pre-experiment questionnaire measurement; (2) Cyberball game [4,37]; (3) number Stroop task and emotional face Stroop task [38,39,40]; (4) post-experiment interview, and (5) questionnaire measurement.

After coming to the laboratory, the participant was told that he/she would complete an interactive online game with two other participants and then complete two computer tasks. In the first session, the participant had to complete pre-experiment questionnaires. In the second session, the participants were asked to play an online Cyberball game with another two confederates. During the Cyberball game, participants were divided into the exclusion or inclusion group and then asked to complete the number Stroop task and emotional face Stroop task sequentially. In the third session, all the participants were instructed to report their feelings during the task (i.e., how did you feel in the Cyberball game) and then filled in post-experiment questionnaires.

### 2.3. Instruments

#### 2.3.1. Cyberball Game

The Cyberball game (see Figure 1) was first developed by Williams and has been updated to version 5.7.0. Before the task, participants were instructed that they would play an online virtual ball-tossing game with two other participants. In addition, the participants were asked to imagine the ball-tossing action to test their mental visualization during the Cyberball game. Participants saw an animated ball-tossing game, and two other players were also in the upper corners of the screen, accompanied by their names. They started with an inclusion condition by having each player receive the ball equally often. After 10 ball tosses, the exclusion group received the ball only once at the beginning but then became ostracized and stopped receiving the ball for the last time in the game. However, the inclusion group received the ball for an equal amount of tosses throughout the game. Both the exclusion and inclusion Cyberball games consisted of a total of 60 ball tosses and lasted approximately 5 min.

#### 2.3.2. Number Stroop Task

In the number Stroop task (see Figure 2), participants needed to count the number of the two to four centrally displayed characters while inhibiting the numerical value of digit characters. Participants were instructed to press the keys 2, 3, or 4 on the keyboard with the index, middle, and ring fingers of the left hand, respectively. They were advised to respond as quickly as possible without too many losses of accuracy. In the congruent condition, the numerical value of the digit was consistent with the number of digits (e.g., 22, 333, 4444). In the incongruent condition, the numerical value of the digit was inconsistent with the number of digits (e.g., 222, 33, 444).

In the number Stroop task, each trial begins with the presentation of a fixation cross in the center of the screen for 500 ms. Then, the target stimulus was displayed on the screen until a response or 2000 ms elapsed. Afterward, the response feedback was presented for 500 ms and was then followed by a blank for 1000 ms. The number Stroop task started with a block of 20 practice trials and was then followed by a block of 180 experimental trials, of which 90 trials were congruent and the other 90 trials were incongruent.

#### 2.3.3. Emotional Face Stroop Task

In the emotional face Stroop task (see Figure 2), participants were asked to identify the facial expression to indicate whether the face stimulus was happy or angry in emotion while ignoring the emotional word (“高兴” means happy, “愤怒” means anger). Participants were instructed to press the keys 3 or 4 on the keyboard with the index and middle of the left hand, respectively. They were advised to respond as quickly as possible without too many losses of accuracy. In the congruent condition, the facial expression was consistent with the emotional words. In the incongruent condition, the facial expression was inconsistent with the emotional words. The trial sequence was the same as for the number Stroop task.

#### 2.3.4. Self-Reported Assessment

Based on the consideration of the practice effect and potential doubt of the Cyberball game, we used different questionnaires before and after the experiment, respectively. Prior to the Cyberball game, each participant was instructed to complete three questionnaires, including a self-esteem scale (SES) to measure individual self-esteem [41], a state-trait anxiety inventory (STAI) for testing trait and state anxiety [42], and a self-rating depression scale (SDS) to assess the level of depression [43]. After all the behavior experiments, each participant was asked to report their feelings during the task verbally and then filled in two questionnaires, including a positive and negative affect schedule (PANAS) to measure both the positive and negative affect [44], and a need-threat scale (NTS) to ensure the success of exclusion manipulation [45].

### 2.4. Statistical Analysis

All data were cast into a 2 × 2 mixed-factor repeated-measures analysis of variance (ANOVA) with congruency (congruent vs. incongruent) as within-subject factors and the group (exclusion vs. inclusion) as between-subjects factors. In addition, the Bonferroni post hoc analysis was undertaken to compare the pairs. A significance level of A = 0.05 was adopted, and all data were analyzed in the statistical package for Social Science (SPSS), version 24.0 (SPSS Inc., Chicago, IL, USA). The Stroop task measurement parameter varied according to the type of task, the means of accuracy (ACC), the means of correct reaction times (RT), and Stroop interference (the difference in RTs between the incongruent and congruent items) which were obtained for each participant, for each condition. Descriptive statistics of the ACC and RT were organized and examined. RT data that were more than 1000 ms or less than 200 ms were excluded from further analysis.

## 3. Results

### 3.1. Subjective-Rating Data

An independent t-test was conducted to determine whether there was a difference between the five questionnaire scores for the exclusion and inclusion groups. As shown in Table 1, the results showed no significant difference in SDS, STAI, and SES scores. However, the need threat scale scores were significantly higher in the exclusion group than in the inclusion group. In addition, compared with the control group, participants in the rejection group reported higher levels of negative affect and lower levels of positive affect in PANAS scores.

### 3.2. Number Stroop Task

With reference to the number of Stroop tasks, both ACC and RT were considered, and Table 2 shows the means and standard deviations of each condition for the exclusion and inclusion groups, respectively. The ACC of all subjects was higher than 90%. There was no significant difference in the accuracy between the exclusion and inclusion groups, so it was not included in further statistical analysis.

The group showed a significant main effect, F (1,60) = 9.36, *p* < 0.01, pη^2^ = 0.23, with longer RTs for the exclusion group than for the inclusion group. The main effect of congruency also reached significance, F (1,60) = 21.76, *p* < 0.001, pη^2^ = 0.44, with longer RTs for incongruency trials than that for congruency trials. The group × congruency interaction was not significant, F (1,60) = 7.21, *p* = 0.45. In addition, the Stroop interference of the exclusion group was significantly higher than for that of the inclusion group, F (1,60) = 15.21, *p* < 0.001, pη^2^ = 0.42, indicating that the exclusion group showed a decreasing inhibitory control in the number Stroop task.

### 3.3. Emotional Face Stroop Task

With reference to the emotional face Stroop task, both ACC and RT were considered, and Table 3 shows the means and standard deviations of each condition for the exclusion and inclusion groups, respectively. The ACC of all subjects was higher than 90%. There was no significant difference in the accuracy between the exclusion and inclusion groups, so it was not in further statistical analysis.

The group showed a significant main effect, F (1,60) = 9.36, *p* < 0.05, pη^2^ = 0.21, with longer RTs for the exclusion group than that for the inclusion group. The main effect of congruency also reached significance, F (1,60) = 11.76, *p* < 0.001, pη^2^ = 0.41, with longer RTs for incongruency trials than that for congruency trials. The group × congruency interaction was not significant, F (1,60) = 9.65, *p* = 0.36. In addition, for the statistical analysis of the Stroop interference, there was no significant difference between the exclusion and inclusion groups.

## 4. Discussion

The purpose of the present study was to investigate social exclusion influences in cool and hot inhibitory control performances. To this end, the participants performed a social exclusion or social inclusion Cyberball game and then completed the cool and hot Stroop task, that is, the number Stroop task and emotional face Stroop task. Replicating previous behavior findings, subjective rating presented multiple consequences of social exclusion [10,45]. The Cyberball game increased the feelings of need-threat and negative affect and decreased the positive affect in the excluded group. The behavior results showed that the RTs of the exclusion group was significantly longer than that of the inclusion group in both cool and hot Stroop tasks, in which the exclusion group showed a significant decrease in processing speed. Notably, in the number Stroop task, the Stroop interference of the exclusion group was significantly higher than that of the inclusion group, but this phenomenon was not shown in the emotional face Stroop task. We will discuss them separately next.

The results of the number Stroop task found significantly longer RTs in the exclusion group than in the inclusion group for both the congruent and incongruent conditions. It demonstrated that social exclusion impairs the cognitive processing speed in the cool inhibitory control. This is consistent with previous studies, proving that social exclusion decreases cognitive control (i.e., reaction time, working memory, self-regulation) [2,8,14] while increasing impulsive behaviors (i.e., aggression, risk-taking) [28,46]. This cognitive ability impairment of social exclusion can be accounted for by the cognitive deconstruction theory, which suggests that social exclusion leads individuals to enter a defensive state of cognitive deconstruction to avoid meaning thought, emotion, and self-awareness [12,28]. This cognitive deconstruction impairs individuals’ ability to self-regulate, leading to more aggressive and impulsive behaviors. Thus, cognitive ability would be impaired after the social exclusion, which was observed in the processing speed, executive function, and reasoning [47]. In the present number Stroop task, the exclusion group required longer reaction times due to the decrease in cognitive ability. In addition, we also found that the Stroop interference in the exclusion group was higher than that in the inclusion group. Numerous studies have shown that negative emotion reduces the ability of inhibitory control [17,18,19,23]. Social exclusion induced varied negative emotions, and it was more difficult to inhibit the conflict between the numerical value and number in the Stroop task. Individuals need to recruit more cognitive resources to suppress the interference of the numerical value of digit characters. To sum up, the number Stroop task found that when experiencing social exclusion, an individual’s positive affective decreased, and the need for threat-related negative emotion increased, resulting in a decrease in the cognitive processing speed and an impaired cool, inhibitory control ability.

For the results of the emotional face Stroop task, we found that the influence of social exclusion on the processing speed, similar to the number Stroop task, decreased in terms of reaction time. However, it is worth noting that the Stroop interference in the emotional face Stroop task did not show a difference between the exclusion and inclusion groups, which was significant compared to the difference in the number Stroop task. Considering the social monitoring system theory, individuals are more sensitive to social information-related stimuli after being socially excluded [29]. Emotional faces, which are a typical form of social information, could capture more attention than emotional words in individuals who experienced social exclusion. Therefore, even though social exclusion weakens an individual’s speed of cognitive processing, the interference effect of words on faces was not affected [40]. This non-Stroop interference could be explained by the sociometer theory, which holds that social exclusion increases the awareness of the relational value and social information to adjust self-esteem [32]. Thus, individuals spend a long time and more cognitive resources to achieve the desired performance outcome. To be specific, socially excluded individuals could allocate sufficient resources to sustain response preparation to social emotion, especially for facial expression [48,49]. In the context of social exclusion, more cognitive resources were allocated to stimuli related to social information, which promoted attention to facial emotion, and the interference of emotional words was relatively weakened. In summary, social exclusion promotes individuals to seek social reconnection and pay more attention to facial expressions rather than emotional words, resulting in a decrease in cognitive processing speed, whereas the hot inhibitory control ability was not changed.

Taken together, in the results of the cool and hot Stroop task, there were two major findings. Firstly, being socially excluded by peers impairs cognitive processes of inhibitory control and leads to longer reaction times in the Stroop task. Secondly, social exclusion impaired cool inhibitory control only in college students and had no significant effect on the hot inhibitory control. According to the cognitive deconstruction theory, social exclusion indeed impaired the cognitive processing speed during the Stroop task. However, through the social monitoring system, they require varieties of cognitive resource allocation to manage emotional processing; socially excluded individuals would pay more attention to social information, especially emotional faces, and thus need to recruit more resources to process facial expressions and ignore emotional words. For this reason, social exclusion facilitates the recognition of facial expressions and reduces the Stroop interference in the hot inhibitory control task.

Consequently, converging with the findings in previous studies, the present two experiments provide us with a comprehensive understanding of how social exclusion influences inhibitory control. It offered a new direction to investigate the interaction between inhibitory control and healthy behaviors during social exclusion. To the best of our knowledge, we are the first to investigate the influence of social exclusion on both the cool and hot inhibitory control. Combined with previous controversies, social exclusion has different effects on an individual’s cool and hot inhibitory control, which may be an important factor leading to the inconsistency between different theories or models [5,50,51]. It is necessary to consider the cool and hot inhibitory control related to factors in promoting prosocial behaviors during social exclusion. For example, a socially excluded individual’s desire to regain acceptance may be sensitive to facial expression. However, when feeling more negative, with facial expression feedback and little possibility of acceptance, they become antisocial. Therefore, if people have experienced social exclusion, they need to have more positive emotional feedback through facial expressions rather than words or other ways. In addition, both social exclusion and inhibitory control cost lots of cognitive resources and would lead to impulsive behavior in studying and working, so it is necessary to avoid conflict during social exclusion and thus reduce irrational behavior.

Finally, the current study still has some limitations that should be addressed. First, only self-reports were used, and no physiological records were used to measure the participants’ emotional responses. Future research is needed to confirm that the social exclusion environment does indeed produce exclusion effects on individuals and the Stroop interferences of the cool and hot inhibitory control are caused by social exclusion. Secondly, inhibitory control is a sub-component of an executive function, which cannot reflect cognitive abilities such as working memory and cognitive flexibility. Future studies are encouraged to evaluate whether other subdomains of an executive function might modulate the impact of social exclusion on inhibitory control, e.g., working memory and cognitive flexibility.

## 5. Conclusions

The current study shows that social exclusion reduces cognitive processing speed and decreases the response speed in both cool and hot inhibitory control. However, only the cool inhibitory control is impaired in socially excluded individuals; hot inhibitory control is not significantly altered.

## Figures and Tables

**Figure 1 ijerph-20-02433-f001:**
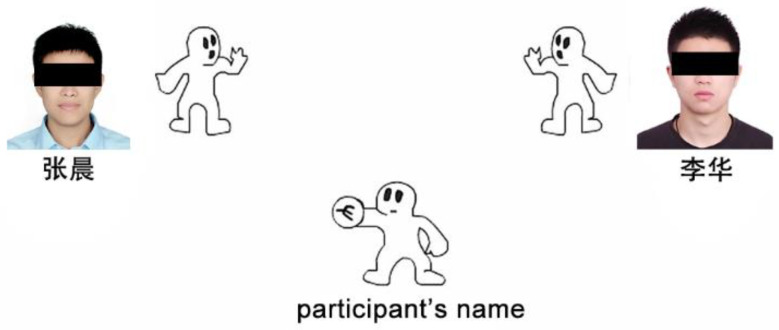
Cyberball game display: Participants are displayed as photographs at the bottom of the screen and the two other players as photographs in the upper corners of the screen. “张晨” means the Chinese name of Zhang Chen; “李华” means the Chinese name of Li Hua.

**Figure 2 ijerph-20-02433-f002:**
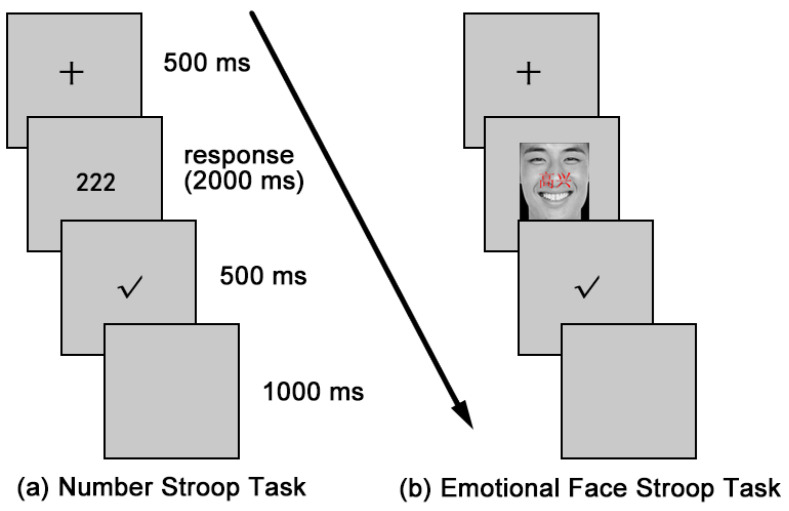
Schematic of trial sequences in cool and hot Stroop task. (**a**) Number Stroop task; (**b**) Emotional Face Stroop task. The words “高兴” in emotional face Stroop task means happy.

**Table 1 ijerph-20-02433-t001:** Means (and standard deviations) of subjective rating in exclusion and inclusion group.

Variable	Exclusion (n = 32)	Inclusion (n = 32)	*p*-Value
SDS	35.04 (7.12)	36.34 (7.88)	0.76
S-AI	37.37 (5.27)	35.36 (5.99)	0.31
T-AI	35.72 (4.64)	35.20 (5.34)	0.38
SES	16.47 (4.22)	17.35 (4.79)	0.59
NTS	31.69 (8.13)	21.75 (6.25)	<0.001
PAS	23.66 (7.88)	27.35 (6.73)	<0.005
NAS	22.35 (5.91)	14.67 (4.57)	<0.005

SDS: self-rating depression scale. S-AI: state anxiety inventory. T-AI: trait anxiety inventory. SES: self-esteem scale. NTS: need-threat scale. PAS: positive affect schedule. NAS: negative affect schedule.

**Table 2 ijerph-20-02433-t002:** Mean (and standard deviations) of the RTs (ms) and ACCs (%) for the cool Stroop task.

Group	ACC (%)		RT (ms)		Stroop Interference
	Congruent	Incongruent	Congruent	Incongruent	
exclusion	97.7 (3.5)	94.4 (3.8)	669.54 (112.83)	759.14 (137.35)	89.6 (24.52)
inclusion	96.8 (2.4)	95.6 (3.2)	639.25 (94.10)	692.42 (126.90)	53.17 (17.81)

**Table 3 ijerph-20-02433-t003:** Mean (and standard deviations) of the RTs (ms) and ACCs (%) for the hot Stroop task.

Group	ACC (%)		RT (ms)		Stroop Interference
	Congruent	Incongruent	Congruent	Incongruent	
exclusion	97.7 (3.5)	93.3 (4.2)	647.32 (82.83)	698.78 (117.35)	51.46 (23.14)
inclusion	98.9 (2.8)	94.4 (3.9)	609.28 (84.10)	657.46 (106.90)	48.18 (12.23)

## Data Availability

The data presented in this study are available upon request from the corresponding author.

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
