# Peer review of "Influence of Social Exclusion on Cool and Hot Inhibitory Control in Chinese College Students"

_ijerph, 2023, doi:10.3390/ijerph20032433_

Round 1

Reviewer 1 Report

1.     The number of subjects is confused. It started with 44 recruits, 32 of them male. They were randomly divided into two groups of 32, 14 of whom were male(2.1).It's totally inconsistent. Please revised it.

2.     The subjects were divided into 2 groups. What were the criteria for this grouping? Before grouping, were the subjects homogeneous or heterogeneous?

3.     As for the detection of emotion, why not conduct the pre-test of emotion before the experiment to determine the baseline level of emotion of each subject?

Author Response

We very thank the reviewer for the comments and suggestions concerning the manuscript. The major revision was on the methods and the results section. Following the reviewer’s suggestion, we we revised the description of participants, the criteria for this grouping, and the experiment manipulation. In addition, the reviewer pointed out there are some grammatical and phrasing error in the manuscript, so we went over the entire manuscript and revised them. Here we detail our responses to the reviewer’s points.

  1. The number of subjects is confused. It started with 44 recruits, 32 of them male. They were randomly divided into two groups of 32, 14 of whom were male(2.1).It's totally inconsistent. Please revised it.

Response: Many thanks for your careful examination. We are so sorry for the carelessness, in our experiments, 44 participants were recruited originally, and 20 more participants were recruited additionally, so there were 64 individuals in the present study. We have revised this sentence, please see the “Track Changes” part in 2.1. 

  1. The subjects were divided into 2 groups. What were the criteria for this grouping? Before grouping, were the subjects homogeneous or heterogeneous?

Response: To help prevent potential doubt of the Cyberball game, rejection sensitivity questionnaire was not used as the criteria for grouping. So instead, participants were grouped by the rejection sensitivity related scales, including Self-esteem Scale, State-trait Anxiety Inventory and Self-rating Depression Scale. The two groups of subjects have been ensure to achieved homogeneous by controlling the self-esteem, anxiety and depression. We have added this part in the “2.1 Participants”

  1. As for the detection of emotion, why not conduct the pre-test of emotion before the experiment to determine the baseline level of emotion of each subject?

Response: Our description of the Self-reported assessment ambiguous. In the present study, we have used the Self-esteem Scale, State-trait Anxiety Inventory and Self-rating Depression Scale before the experiment. Based on the consideration of the learning effects of testing, we only used the Positive and Negative Affect Schedule after experiment. We have rewritten this part, please see “2.3.4 Self-reported assessment”

Reviewer 2 Report

I recognise the relevance of the object of study of this article. It introduces a new analysis of the effects of social exclusion at the level of reasoning and at the level of learning processing, i.e. it explores the cognitive dimension of the person.

It has a good and appropriate methodology, presented clearly, as well as data collection are sustainable and favor a good discussion of results.

I suggest that the conclusion is strengthened at the level of the relationship of social exclusion with cognition.

The bibliographical references are adequate 

Author Response

We very thank the reviewer for the comments and suggestions concerning the manuscript.  Following the reviewer’s suggestion, we added some references and revised some conclusions.

Reviewer 3 Report

The authors examined the effects of social exclusion on cool and hot inhibitory control. Overall, the manuscript is well written, and I have a few minor comments/suggestions.

1) 1. Introduction: (a) More elaboration is needed for cool and hot inhibitory control. (b) Are they the same as hot EF and cool EF? (c) Based on the theories and models of social exclusion discussed, what is the hypothesis?

2) 2.3.4 Self-reported assessment: What is their internal consistency in this study?

3) 4. Discussion: It seems that the important finding of this study is that social exclusion has different effects on individuals’ cool and hot inhibitory control. Please discuss more regarding this (e.g., possible explanations for the different effects, their theoretical and/or practical implications, etc.)

4) 4. Discussion: “It is necessary to consider the cool and hot inhibitory control related factors in promoting prosocial behaviors during social exclusion.” Kindly give several examples of these factors.

Author Response

We very thank the reviewer for the comments and suggestions concerning the manuscript. The major revision was on the introduction and discussion section. Following the reviewer’s suggestion, we we revised the description of cool and hot inhibitory, the experiment manipulation and the discussion of varied results. Here we detail our responses to the reviewer’s points in this attachment.

  1. Introduction: (a) More elaboration is needed for cool and hot inhibitory control. (b) Are they the same as hot EF and cool EF? (c) Based on the theories and models of social exclusion discussed, what is the hypothesis?

Response: Following your suggestion, we have provided more background in the introduction. (a) Many thanks for your suggestion, we have added more elaboration in introduction, please see in Line 65-69. (b) We have added the description of the relationship between inhibitory control and executive function, please see in Line 77-82. (c) We have added the hypothesis in the present experiment: “Following the above theories and models of social exclusion, we hypothesized that social exclusion facilitates hot inhibitory control in emotional Stroop task, but impairs cool inhibitory control in non-emotional Stroop task.” Please see in Line 108-111.

“Here, we investigate one appropriate candidate to measure the interaction, namely executive function (EF), which has three factors in measurement, including inhibitory controlm, working memory and cognitive flexbility. Zelazo and Carlson have divided EF into two aspects, hot EF and cool EF[15, 16]. Even though both hot and cool EF are top-down processing that depends on cognitive effort, they require varies of mental resource to manage emotional processing. .........For instance, one study reported that social evaluation facilitated hot inhibitory control in adolescents, but this facilitation effect was not found in cool inhibitory control [22]. Another recent study suggested that cool and hot inhibitory control performance were not correlated adolescents [23]. Taken together these findings suggest that hot and cool inhibitory control depend on different cognitive and emotional processing, which are similar to hot and cool EF.”

  1. 3.4 Self-reported assessment: What is their internal consistency in this study?

Response: To help prevent potential doubt of the Cyberball game, rejection sensitivity questionnaire was not used as the criteria for grouping. But instead, participants were grouped by the rejection sensitivity related scales, including Self-esteem Scale, State-trait Anxiety Inventory and Self-rating Depression Scale. The two groups of subjects have been ensure to achieved homogeneous by controlling the self-esteem, anxiety and depression. The 3.1 Subjective-rating data showed no significant difference in SDS, STAI, and SES scores between two groups. It supported that participants are homogeneous in the experiment.

  1. Discussion: It seems that the important finding of this study is that social exclusion has different effects on individuals’ cool and hot inhibitory control. Please discuss more regarding this (e.g., possible explanations for the different effects, their theoretical and/or practical implications, etc.)

Response: Many thanks for your suggestion, we have expanded on the explanations for different results in discussion section, please see Line 318-329.

“Taken together the results of cool and hot Stroop task, there were two major finding. Firstly, socially excluded by peers impairs cognitive processes of inhibitory control, leads to longer reaction time in Stroop task. Secondly, social exclusion impaired cool inhibitory control only in college students, and had no significant effect on the hot inhibitory control. According to the cognitive deconstruction theory, social exclusion indeed impairs cognitive processing speed during Stroop task. However, by the social monitoring system, they require varies of cognitive resource allocation to manage emotional processing, socially excluded individuals would pay more attention to social information, especially emotional faces, and thus need to recruit more resources to processing facial expression and ignore the emotional words. For this reason, social exclusion facilitates the recognition of facial expression and reduce the Stroop interference in hot inhibitory control task.”

  1. Discussion: “It is necessary to consider the cool and hot inhibitory control related factors in promoting prosocial behaviors during social exclusion.” Kindly give several examples of these factors.

Many thanks for your suggestion, we have expanded on examples involving inhibitory control prosocial behaviors during social exclusion in discussion section. Please see Line 339-347.

 “For example, socially excluded individuals desire to regain acceptance and thus may sensitive to facial expression. However,when feeling more negative facial expression feedback and little possibility of acceptance, they become antisocial. Therefore, if people have experienced social exclusion, they need to have more positive emotional feedback by facial expression rather than words or other ways. In addition, both social exclusion and inhibitory control costs lots of cognitive resources and would leads to impulsive behavior in studying and working, so it is necessary to avoid conflict during social exclusion and thus reduce irrational behavior.”

Round 2

Reviewer 1 Report

accept the manuscript for publication.